# Feasibility of Radar Vital Sign Monitoring Using Multiple Range Bin Selection

**DOI:** 10.3390/s25082596

**Published:** 2025-04-20

**Authors:** Benedek Szmola, Lars Hornig, Karen Insa Wolf, Andreas Radeloff, Karsten Witt, Birger Kollmeier

**Affiliations:** 1Department of Neurology, School of Medicine and Health Science, Carl von Ossietzky Universität Oldenburg, 26129 Oldenburg, Germany; karsten.witt@uni-oldenburg.de; 2Medizinische Physik, Carl von Ossietzky Universität Oldenburg, 26129 Oldenburg, Germany; birger.kollmeier@uni-oldenburg.de; 3Fraunhofer Institute for Digital Media Technology IDMT, Oldenburg Branch for Hearing, Speech and Audio Technology HSA, Marie-Curie-Straße 2, 26129 Oldenburg, Germany; lars.hornig@idmt.fraunhofer.de (L.H.); karen.insa.wolf@idmt.fraunhofer.de (K.I.W.); 4Division of Otolaryngology, Head and Neck Surgery, Carl von Ossietzky Universität Oldenburg, 26129 Oldenburg, Germany; andreas.radeloff@uni-oldenburg.de

**Keywords:** radar, vital sign monitoring, sleep monitoring, lateral placement, multiple range bin selection, topological data analysis, persistence diagram, time delay embedding, chirp median

## Abstract

Radars are promising tools for contactless vital sign monitoring. As a screening device, radars could supplement polysomnography, the gold standard in sleep medicine. When the radar is placed lateral to the person, vital signs can be extracted simultaneously from multiple body parts. Here, we present a method to select every available breathing and heartbeat signal, instead of selecting only one optimal signal. Using multiple concurrent signals can enhance vital rate robustness and accuracy. We built an algorithm based on persistence diagrams, a modern tool for time series analysis from the field of topological data analysis. Multiple criteria were evaluated on the persistence diagrams to detect breathing and heartbeat signals. We tested the feasibility of the method on simultaneous overnight radar and polysomnography recordings from six healthy participants. Compared against single bin selection, multiple selection lead to improved accuracy for both breathing (mean absolute error: 0.29 vs. 0.20 breaths per minute) and heart rate (mean absolute error: 1.97 vs. 0.66 beats per minute). Additionally, fewer artifactual segments were selected. Furthermore, the distribution of chosen vital signs along the body aligned with basic physiological assumptions. In conclusion, contactless vital sign monitoring could benefit from the improved accuracy achieved by multiple selection. The distribution of vital signs along the body could provide additional information for sleep monitoring.

## 1. Introduction

Sufficient good quality sleep is one of the pillars of overall human health. While many people know about the importance of sleep [1], sleep deprivation is still common in adults [2,3]. Impaired sleep has been linked to various adverse health outcomes, such as an increased risk of cardiovascular disease, diabetes and a decline in cognitive performance [1,2,4,5]. Furthermore, because a lack of good quality sleep leads to daytime sleepiness and fatigue, it is associated with an elevated risk of suffering traffic accidents as well as occupational accidents [6,7].

Sleep analysis can shed light on various sleep disorders, which can in turn be targeted with therapies. The gold standard sleep analysis method is polysomnography (PSG) [8,9]. PSG is a multimodal system where bodily signals relevant for sleep are recorded. The following are important recorded signals: electroencephalogram (EEG), electrooculogram (EOG), electrocardiogram (ECG), electromyogram (EMG) of chin and leg muscles, airflow, respiratory effort and oxygen saturation. The standard method of conducting a PSG measurement is performed in a dedicated sleep laboratory, where a trained staff member watches over the patient the whole night and controls the measurement. Then, an expert sleep scorer annotates the whole recording, marking sleep stages and respiratory events among others [10].

While PSG is a highly sensitive tool for analyzing sleep, its complex nature also has a downside. Because of the staff and location requirements, access to PSG for patients is limited, and the cost can also be a limiting factor [11,12]. Furthermore, the so-called first-night effect can present a problem in the interpretation of results [13,14]. This effect refers to patients sleeping worse because of the foreign environment or the attached sensors. Back to back nights with PSG might be a solution, although this leads to further costs and logistical complexity. Furthermore, it also has been suggested that the follow up nights are not representative of normal sleep; rather, the patient is in a recovery phase after the initial poor sleep [13].

There are simplified versions of PSG, which can be performed at home. These are collectively referred to as Home Sleep Tests (HST) [15]. While HST has the advantage of being performed in a familiar environment, the disturbance caused by the contact sensors is still present. Furthermore, the costs of the whole analysis and the logistics of delivering the sensors to the patients’ homes are still limiting factors. Alternatively, if the patients bring the sensors home from a hospital, there is the risk of improper setup of the sensors.

Given the challenges associated with PSG, alternative sleep analysis techniques have been suggested. There are approaches with both contact-based sensors and sensors requiring no body contact [16,17,18]. These contact-based sensors largely mirror sensors used in PSG, albeit in different configurations, using a single sensor or a small number of sensors [16]. Sensors placed on the mattress represent an in-between solution as they are not technically contactless but are not intrusive like devices attached to the body [19]. Examples for contactless sensors which have been investigated for sleep monitoring include ambient microphones [20], infrared cameras [21] and radar [22].

Biomedical radars are receiving increasing attention in the sleep monitoring research literature and vital sign monitoring more broadly [23]. A radar system provides a convenient, non-contact way to measure macro and micro body motions such as limb movements, breathing and heartbeats [24,25]. Compared to other contactless sensors, radars generate less privacy-sensitive data. The setup for sleep monitoring only requires one to place the device near the bed and orient it towards the sleeping person. Given its ease of application, long term monitoring is easily realizable, both at home or in a clinical setting. For sleep monitoring, accurate long term measurement is highly valuable, as it can provide insights on variations in sleep quality over longer time frames. Because of the above described limitations, such insights are not gained in PSG-based sleep analysis.

Most studies use one of three radar types: continuous wave (CW) doppler [26], frequency modulated continuous wave (FMCW) [22,27,28,29,30] and impulse radio ultra wideband (IR-UWB) [31] radar. FMCW radars provide a good balance of complexity and performance. By modulating the frequency of transmitted waves, targets located at various ranges in front of the device can be distinguished. The individual spatial segments which the radar can distinguish are referred to as range bins. Previous works have not taken full advantage of this feature of FMCW radars in the context of sleep vital sign monitoring, as they had the device aimed straight at the chest or back of the person. In our previous publication [30], we presented the viability of lateral radar placement relative to the person (e.g., at the foot end of the bed or on the nightstand), where the body parts are mapped onto separate range bins (illustrated in Figure 1). As a result, body motions can be extracted from different body regions simultaneously. One advantage is the extraction of small-amplitude heartbeats from range bins in which the comparatively large breathing motion is not masking them. For example, the legs could be a suitable region for detecting heartbeats, as breathing motion is not present there. Figure 2 shows an exemplary radar signal, where breathing motion is visible over multiple range bins.

Biomedical time series, such as breathing and heartbeat, are non-stationary signals, as both amplitude and frequency change over time. Classical signal processing methods become limited in their effectiveness when applied to such signals. In contrast, persistence homology [32]—an algorithm from the field of topological data analysis (TDA) [33]—is well suited for analyzing non-stationary periodic time series [34]. Persistence homology summarizes information on the shape of the data. For example, in a doughnut-shaped point cloud, persistence homology could show how densely packed the individual points are, and how large the hole of the doughnut is. Two common methods of computing persistence homology of a time series signal are sublevel set filtration directly from the signal or Vietoris–Rips filtration after time delay where the signal is embedded first [33,35]. Time delay embedding after Takens’ theorem [36] is a technique where a one-dimensional time series is transformed to a multi-dimensional phase space by choice of appropriate embedding parameters [37]. Skaf and Laubenbacher [38] have provided an overview on the use of TDA in various fields of biomedicine. Chung et al. (2021) have used TDA for sleep staging based on heart rate variability computed from ECG [35]. Chung et al. (2024) used TDA for sleep staging using airflow recordings [34]. Using persistence homology, Erden and Cetin [39] estimated the respiratory rate of participants recorded with a non-contact pyroelectric infrared sensor.

The main aim of this article is to present a method which takes full advantage of the novel placement strategy for FMCW radars [30]. With this placement, the body is mapped onto multiple range bins to reduce the overlap of different body motions in the radar signal. We propose a method based on persistence homology that selects every range bin containing breathing or heartbeat signals for vital rate computation. The result is the distribution of vital signs along the body over time. Since, during rest or sleeping, breathing is the dominant movement along the upper body, we expect the algorithm to find breaths in this region, whereas we expect heartbeats to be preferentially found along the legs, where breathing motion is absent. Computing vital rates from multiple range bins should lead to higher accuracy, because more information about the same underlying phenomenon is utilized. We present a comparison between our previous results using single bin vital rate computation and the new proposed method. The proposed method shows a clear improvement in terms of accuracy to the reference sensors.

In the following, we introduce the FMCW radar measurement principle (Section 2.1) and the usage of the median for enhancing radar signal clarity (Section 2.2). Then, we describe the multiple bin selection algorithm based on persistence homology (Section 2.3), followed by the methods for deriving breathing and heart rate (Section 2.5). Next, we describe the range bin distribution produced by the multiple bin selection (Section 3.1), and compare the vital rate accuracy from the proposed method to previous results from single bin selection (Section 3.3). Finally, we discuss the potential uses of the distribution of breathing and heartbeat signals over time (Section 4).

## 2. Materials and Methods

### 2.1. FMCW Radar Measurement Principle

In this work, we used data from an FMCW radar. This section gives an overview of the working principle of the FMCW radar and the signal preprocessing steps that we used. For a more detailed introduction to radar vital sign monitoring, see the book by Boric-Lubecke et al. [24] and the review by Paterniani et al. [25].

FMCW radars operate by emitting a continuous electromagnetic wave at the carrier frequency $f_{c}$ with periodic frequency modulation. We used a 60 GHz model (IWR6843ISK-ODS) from Texas Instruments (Dallas, TX, USA). This device works with a linear ramp, thereby creating linear frequency sweeps, called chirps. Chirps can be characterized by their bandwidth and slope. The bandwidth *B* of the frequency sweep defines the range resolution ΔR of the range bins(1)$$\Delta R=\frac{c}{2B}\;,$$
where *c* is the speed of light. The device we used was set to have a bandwidth of 3 GHz, and therefore the range resolution was 5 cm. The slope of the chirp μ can be expressed as(2)$$\mu =\frac{B}{T}\;,$$
where *T* is the duration of the frequency sweep. The distance to the monitored person $d_{subj}$ can be determined by comparing the frequency of the transmitted and the received signals, as the frequency ramp will be delayed by τ proportionally to the distance the received wave has travelled (time of flight principle). The frequency difference Δf is(3)Δf=μτ,
and the distance to the person can be derived using the formula(4)$$d_{subj}=\frac{\tau c}{2}\;.$$

The received signal is mixed with the transmitted signal on one channel (in-phase, I channel) and also with a 90 degree phase-shifted copy of the transmitted signal on another channel (quadrature, Q channel). The combination of the I and Q channels is the complex-valued intermediate frequency (IF) signal. After mixing, a low-pass filter is applied to the IF signal to remove high frequency components. Given the short-range application context [40], the IF signal $x_{IF}\left(t\right)$ can be expressed as(5)$$x_{IF}\left(t\right)=Aexp\left(j\left(2\pi \Delta ft+\phi (t)\right)\right)\;,$$
where A is the signal amplitude and ϕ(t) is the time varying phase, the latter of which is influenced by the small body motions, $d_{subj}\left(t\right)$, such as breathing and heartbeats(6)$$\phi \left(t\right)=\frac{4\pi \left(d_{subj}\left(t\right)+d_{subj}\left(t_{0}\right)\right)}{\lambda }\;.$$

To extract the magnitude and phase information from the complex IF signal, we use the Fast Fourier Transform (FFT). To reduce spectral leakage, we use a Hanning window before applying the FFT. Computing the FFT along one chirp returns what is called the range FFT. By stacking the range FFTs for each chirp after each other, we create a range–time matrix. Here, the rows represent the range bins and the columns represent the consecutive chirps across time. Computing the magnitude along one column gives a snapshot estimate of the reflected signal strength across range bins. When a magnitude time series is built from consecutive snapshots, the change over time in reflection magnitude can be analyzed, which can help to distinguish between static and moving objects [29]. By extracting the phase of a range bin across time instead of magnitude, we can observe the micro-movements detected by the radar within that segment of space. Multiple authors have investigated the phase-based distance measurement accuracy of FMCW radars, including 60 GHz models [41,42,43]. They reported phase-based accuracy in the µm range within one range bin. For extracting the phase information, we used arctangent demodulation followed by phase unwrapping to avoid phase discontinuities.

The radar used here was configured to transmit 12 chirps in rapid succession every 50 ms [30]. We refer to one set of 12 chirps as one frame. Consecutive chirps can be used to obtain velocity information on the objects recorded by the radar, but in this work, we did not perform such analysis. Instead, as detailed in the following section, we utilized the chirps of one frame to improve signal quality.

### 2.2. Chirp Median

Before data analysis, we visually explored the radar dataset, aided by the reference PSG signals. Some time windows showed radar heartbeats barely distinguishable from noise. We found that taking the median over chirps in a frame lead to clearer heartbeat signals. Here, we discuss why summarizing the chirps is applicable when analyzing breathing and heartbeat signals.

The time between the start of consecutive chirps was 401 µs. In one frame, the time elapsed from the start of the first chirp to the end of the last chirp was 4.812 ms. This duration is approximately 1/400 of the period of fast breathing (30 breaths per minute) [44,45], and 1/60 of the period of the theoretical maximal heart rate of a 20 year old (200 beats per minute) [46,47]. Therefore, we can safely apply the median across the chirps, without losing information about the vital signs. By taking the median across the 12 chirps, the random noise present in each chirp’s signal is reduced, while the common signal is preserved. Additionally, as we use the median instead of the mean, the summation of chirps is robust against outliers.

To test whether taking the median of chirps improves radar heartbeat detection performance, we computed heart rates both with and without the chirp median. When not applying the chirp median, the first chirp from each frame was used. Range bin selection was performed using chirp median in both cases. This way, the rates are computed from the same range bins; the only difference is the chirp median.

### 2.3. Proposed Method

In this section, we provide a brief introduction to the background of the methods used, followed by a description of the proposed multiple range bin selection algorithm. For a more in-depth introduction to the use of persistence homology for time series, please refer to [33,35].

#### 2.3.1. Methodology Background

First, we introduce the notion of simplicial complexes as they are the foundational building blocks of TDA. A simplex can be best understood as a generalization of a triangle to arbitrary dimensions. A 0-simplex is a vertex, a 1-simplex is a line segment, a 2-simplex is a triangle, a 3-simplex is a tetrahedron and so on. A subset of a simplex is called a face, which can be well understood by imagining a tetrahedron, the subsets of which are its triangular faces. A collection of simplices, where each intersection between two of the simplices is a face of both simplex, is called a simplicial complex.

Homology theory is part of algebraic topology, of which we will give a practical introduction necessary for understanding our proposed method. The homology of a data set can be understood as the description of topological features across dimensions. For dimension 0 ($H_{0}$), these features are connected components, for dimension 1 ($H_{1}$), there are holes surrounded by a loop and for dimension 2 ($H_{2}$), there are enclosed voids. Higher dimensions cannot be well imagined. A line segment has one $H_{0}$ feature, a circle has both a $H_{0}$ and $H_{1}$ feature, while a ball has one $H_{0}$ and one $H_{2}$ feature.

The data we are working with are in the form of one-dimensional time series, but the Vietoris–Rips filtration (introduced later) works on point clouds. Therefore, we need a method for transforming time series to point clouds. A well-suited solution is time delay embedding (TDE), which creates a multi-dimensional representation of a time series by using time lagged versions of the original data. Takens’ theorem [36] states, that with appropriate embedding parameters, a good approximation of the underlying dynamical system’s state space can be derived from the observed time series. In the case of a breathing or heartbeat signal, the underlying systems are the complex physiological systems that govern breathing and heartbeat motion which can be recorded using radar.

For a time series x(t), the embedded version x(t,τ,m) is(7)$$x\left(t,\tau ,m\right)=\left(x(t),x(t-\tau ),\ldots ,x(t-(m-1)\tau )\right)\;,$$
where τ is the delay and *m* is the dimension of the embedding. We used this so-called uniform version of TDE. For an introduction to various embedding methods, see [37]. Panel B in Figure 3 shows the result of applying TDE to a radar breathing time series (panel A). From the one time series, two additional time series are created by shifting 10 and 20 samples, respectively. When the three time series are plotted together, they form a three-dimensional point cloud arranged in a loop formation. This form is characteristic of periodic signals. Referring back to the importance of embedding parameters, with an inappropriate choice (e.g., τ=1), the loop would collapse into a straight line.

Having transformed the one-dimensional time series to a point cloud, we can introduce Vietoris–Rips complexes. Given a three-dimensional point cloud *S* with points $v_{i}\in R^{3}$ and a distance function $dist(v_{i},v_{j})$, a Vietoris–Rips complex is a set of points T(r)⊂S, where(8)$$\forall v_{i},v_{j}\in T\left(r\right),\;dist\left(v_{i},v_{j}\right)<2r\;.$$

For the purposes of this work, $dist(v_{i},v_{j})$ is the Euclidean distance function, but other functions could be used as well. A good intuition for a Vietoris–Rips complex is to imagine spheres with radius *r* around each member of the point cloud. Those members whose spheres touch belong to the Vietoris–Rips complex T(r). A Vietoris–Rips filtration is a series of subcomplexes each with increasing radius. The first subcomplex in the series, when r=0, is just the separate points of the point cloud. Then, as the radius grows, multiple points merge into complexes until finally all points belong to the same complex.

Another type of useful filtration is called sublevel set filtration. This method works with functions, which is practical as no transformation is required to apply it to our time series. The sublevel set $L_{r}\left(f\right)$ of a function f(x) is(9)$$L_{r}\left(f\right)=\left\{x:f(x)\leq r\;,x\in R\right\}\;.$$

Any two neighboring samples of the input function $x_{i},x_{i+1}\in L_{r}$ form a 1-simplex, a line segment. Sublevel set filtration can be understood by imagining a line sweeping up along a time series. Consider, for example, the time series shown in panel A of Figure 4. The line starts at y=0 and rises until y=1. Values of interest are marked with dotted lines and the corresponding *y*-values are shown. At any point during the sweep, parts of the time series below the line are simplices. As an example, in panel A of Figure 4, at y=0.37, there are three separate line segments below the line (red segments in Figure 4). When the line passes a critical point of the time series, two simplices will merge together, as the point which was separating them is now also below the sweeping line. One such merge happens at y=0.71 in the example figure (yellow segment in Figure 4).

If we want to apply homology to the Vietoris–Rips complexes or the sublevel sets discussed before, a challenge presents itself. At which scale should the homology be computed? In other words, what value should be set for *r* in Equations (Equation 8) and (Equation 9)? If *r* is too small, no topological features are present, and at the other extreme, if *r* is too large, every point is connected, and the characteristic features are hidden. Persistence homology offers the solution. It tracks the topological features as they evolve with the increasing scale. Features which are less robust, often because they represent noise, will disappear shortly after forming. Robust features, which persist across many scales, carry the most information on the underlying data. The scales at which topological features appear and disappear are referred to as the time of birth $\left(b_{i}\right)$ and death $\left(d_{i}\right)$, respectively.

There are various implementations of persistent homology in different programming languages. In our work, we used the Python implementation “Ripser.py” [48] for computing the Vietoris–Rips and sublevel set filtrations.

The persistence diagram is a summary of the birth–death pairs $\left(b_{i},d_{i}\right)$ of the topological features. The diagram plots the death times against the birth times. Points near the diagonal represent features which disappeared shortly after they were formed, and are most often noise. Points which are far above the diagonal represent robust topological features, which carry information on the source data. The persistence diagram can also be transformed to the birth–lifespan diagram by subtracting from each death value the corresponding birth value. This transformation emphasizes how long each topological feature persisted.

The panel C of Figure 3 shows the persistence diagram that was computed from the previously presented TDE of the example radar breathing signal based on the Vietoris–Rips filtration. The blue points are the $H_{0}$ features. As the points in the point cloud are all densely packed, the diagram does not show any outliers here. On the other hand, the $H_{1}$ points shown in green, have one clear outlier. This outlier point represents the large prominent loop in the point cloud, the one-dimensional topological feature.

An example of applying persistence homology to a sublevel set filtration is presented in panel B of Figure 4. By looking at the dotted lines on the two panels of the figure, the source of the $H_{0}$ features—shown as blue points—can be understood. To give one example, the point at (0.47,1.00) in the diagram corresponds to the first half-wave in the time series. The feature appears at a value of y=0.47, and is merged with the other features at y=1.00.

#### 2.3.2. Multiple Range Bin Selection Method

Here, we describe the proposed multiple range bin selection method which is based on the techniques explained above. We start with the bin selection for detecting breathing, followed by the explanation for the bin selection for detecting heartbeat.

##### Detection of Breathing

Breathing range bin selection has two branches: one uses the Vietoris–Rips filtration (green fields in Figure 5), while the other uses sublevel set filtration (red fields in Figure 5) to derive persistence diagrams. In both cases, the first step is applying a 5 Hz low-pass filter to the data, which rejects the high frequency noise while preserving the detailed breathing waveform.

In the Vietoris–Rips branch, the signal is embedded before the filtration using TDE with the parameters τ=10 samples (0.5 s) and m=3. Then, the $H_{0}$ and $H_{1}$ persistence diagrams are computed using the ripser function from the Ripser library. The diagrams are subsequently converted from birth–death to birth–lifespan representation.

To evaluate the diagrams, we use the Density-Based Spatial Clustering of Applications with Noise (DBSCAN) algorithm [49] to split the $H_{1}$ diagram points into a noise cluster and informative outliers [50]. DBSCAN has two parameters. The first is ε, which sets the maximum distance between two points for them to be considered neighbors. The second parameter is minSamp, the minimum number of points that have to be in a point’s neighborhood for it to be considered a core point. We set minSamp to 2, as we are looking for lone outliers. For ε, we compute a secondary Vietoris–Rips filtration to derive the furthest distance between the diagram points.

If DBSCAN finds outliers, their birth and lifespan values are tested. Birth values must be either close to the maximal $H_{0}$ value, signifying a clear signal where the loop of the embedding forms when all points are connected, or must overlap with the $H_{1}$ noise cluster’s birth distribution to account for more noisy signals, where the loop forms before all points are connected.

Outlier points’ lifespan values are tested for being sufficiently larger than the lifespan of the $H_{1}$ noise cluster and the maximal $H_{0}$ lifespan. Both of these criteria test whether the outlier represents a robust loop in the embedding.

The sublevel set filtration branch also applies the DBSCAN algorithm, in this case on the $H_{0}$ diagram. Different from the Vietoris–Rips case, here we are looking for noise and signal clusters instead of outliers. As the points of the signal cluster represent the individual oscillations of the time series, we set the minSamp parameter to the minimal number of expected breaths (10 breaths per minute, BPM) [44,45]. The ε parameter is derived as described for the Vietoris–Rips filtration branch.

Two criteria are evaluated in the sublevel set branch: The first of these is whether the number of points in the signal cluster are in accordance with the physiological breathing frequency range (10–22 BPM) [44,45]. For the second criterion, the lifespan values of the signal cluster are tested in relation to the noise cluster and the time series amplitude range, to determine the signal clarity.

Both preliminary decisions of the two branches are finally combined. Signals are determined to show breathing either if both branches reached that conclusion or if in one of the branches they passed stricter versions of the same criteria.

##### Detection of Heartbeat

Heartbeat range bin selection only uses sublevel set filtration as the combination with the more complex Vietoris–Rips filtration did not show a benefit. Figure 6 shows a flowchart of the developed algorithm.

The time series are band-pass filtered in the 0.65–5 Hz range, which eliminates breathing and high frequency noise, but preserves the heartbeat waveforms. The $H_{0}$ persistence diagram is computed with sublevel set filtration and subsequently converted to birth–lifespan representation. Then, DBSCAN is applied to the diagram. The minSamp parameter is set to correspond to the lowest expected heart rate, 40 beats per minute (BPM) [46,47]. For ε, the data-driven approach did not work well; therefore, we used an empirical value based on the time series amplitude range.

The criteria for evaluating the diagram are analogous to those used in the breathing sublevel set filtration branch. As there is no second branch for heartbeat bin selection, the result of this branch is the final output.

### 2.4. Exploration of Windows with No Selected Range Bins

Time windows where no range bins were selected by the proposed method were closely inspected. Visual analysis suggested that these windows were often corrupted by artifacts. To quantitatively test this observation, we computed three measures separately in two groups: time windows where the proposed method selected no range bins and time windows where it selected at least one range bin. All three measures were based on the radar magnitude time series.

The first measure is based on the standard deviation. The standard deviation of the magnitude over a time window has been shown in the literature to be a good discriminator between static objects and those that move within the analyzed time window [29]. Larger body motions will lead to larger changes in the magnitude signal, therefore also raising the standard deviation. As we wanted to compare the range bins with highest variance, we took the 90th percentile of all included range bins’ standard deviation values in each time window. Then, for the comparison between the two groups of time windows, we computed the quartiles from all of the extracted values.

The other two measures were aimed at quantifying differences in the signal distribution. In time windows without artifacts, it is assumed that the magnitude time series does not show outliers and is approximately normally distributed. Artifacts change the distribution, which can be detected by computing the skewness and kurtosis. Skewness describes how much a distribution is shifted relative to a symmetric distribution. Kurtosis quantifies the “tailedness” of the distribution, which refers to the prevalence of outliers. In each time window, the average skewness and kurtosis were computed. To compare the group of time windows with no selected range bins, and the group with selections, the following ratios were computed for both groups: ratio of windows where the absolute skewness was above 1, and ratio of windows where the absolute kurtosis was above 3. The two thresholds were chosen empirically based on visual inspection.

### 2.5. Vital Rate Computation

The vital rate computation methods were carried over from our previous publication, as they already showed good performance. Here, we will summarize the key aspects of the algorithms; for a more thorough introduction, we refer the reader to [30].

Breathing rates from both the thorax belt and radar were computed using autocorrelation. By determining the (non-zero) time lag between the signal and itself where the correlation score is highest, the signal’s period can be determined. The radar breathing rate was first computed for every selected range bin. Subsequently, the median was derived over the individual bins’ rates. This is new compared to the original publication [30], in which the breathing rate was derived from only one range bin.

ECG and radar have different signal characteristics; therefore, heart rates were computed using separate algorithms for each signal type [30]. To compute heart rates from ECG, a method based on the stationary wavelet transform (SWT) was used. Specifically, the “Symlet4” wavelet was chosen because of its similarity in shape to the characteristic QRS complex of ECG. Heartbeats were detected using peak finding, and then the heart rate was computed from the peak-to-peak intervals.

Radar heartbeats were also detected using peak finding after a normalization process that was described by Choi et al. [28]. Apart from the signal transformation, the heart rate computation was performed analogously to ECG. As with radar breathing rates, the median of the individual bins’ results was taken as the final output. This last step was modified compared to our previous publication, where the heart rate was computed only from a single range bin.

The radar-derived vital rate error was quantified in relation to PSG using the mean absolute error (MAE) and the mean absolute percent error (MAPE): (10)$$MAE=\frac{1}{N}\sum_{n=1}^{N}\left|f_{PSG}\left(n\right)-f_{Radar}\left(n\right)\right|\;,$$(11)$$MAPE=100\times \frac{1}{N}\sum_{n=1}^{N}\frac{|f_{PSG}\left(n\right)-f_{Radar}\left(n\right)|}{f_{PSG}\left(n\right)}\;,\;$$
where $f_{Radar}\left(n\right)$ and $f_{PSG}\left(n\right)$ are the radar and PSG vital rates of the *n*th time window, respectively.

### 2.6. Description of Dataset

#### 2.6.1. Study Description

We used the dataset from [30], where healthy volunteers were measured overnight. There were three radars set up at different positions surrounding the bed where the participants slept, as the aim of the original study was to validate the usage of the novel lateral radar placement. In this work, we focus on the radar positioned at the foot end of the bed. This radar was positioned 1 m above the top of the mattress and 1.25 m from the middle of the bed, and it was pointed onto the middle of the bed. Reference signals were recorded in parallel using a PSG system. During the recording, participants were not constrained to a certain sleeping posture. The only instruction was to sleep with their feet pointed towards the radar (as illustrated in Figure 1).

#### 2.6.2. Radar Device

For the details of the radar device, see Section 2.1. To record the radar signals and transmit them to the recording laptop, the following parts were used in addition to the radar sensor board itself: a real time measurement board (DCA1000EVM, Texas Instruments) and a Raspberry Pi 4B for wireless data transmission. A custom made box was used to house these three components. For saving and synchronizing the data from the three radar devices, the Lab Streaming Layer (LSL) protocol [51] was used. Synchronizing the PSG and radars was carried out manually, based on specific leg motions performed by the participants as a marker pattern.

#### 2.6.3. Reference Device

Reference signals were recorded with the PSG device “SOMNOscreen plus” from SOMNOmedics (Randersacker, Bavaria, Germany). This work used the piezoresistive thorax belt for breathing reference and the ECG for heartbeat reference.

#### 2.6.4. Participants

The inclusion criteria for the participants were to be healthy adults capable of consenting to the study. Overall, eleven participants took part in the study (four females, seven males, mean age: 32.7 years, range: 19–45 years). Five recordings were excluded from the data pool, two because of sensor failure and three because of failure to synchronize the radar and reference systems. Therefore, after exclusions, the recordings from six participants were used. Every participant provided written informed consent. Review and approval of the study was given by the Commission for Research Impact Assessment and Ethics of the Carl von Ossietzky Universität Oldenburg (protocol number: Drs.EK/2021/079-01, date of approval: 7 September 2022).

### 2.7. Epoching and Range Bins of Interest

We conducted our analysis using a sliding window method. The step between the start of consecutive windows was 5 s. For range bin selection, the windows had a length of 15 s. After the range bin selection step, the 15 s windows were merged into 60 s. Vital rate computation was performed in the 60 s windows to have a high frequency resolution. The 60 s windows were created by checking the 15 s windows for continuity on a bin by bin basis. If over a span of 10 consecutive 15 s windows a range bin was selected at least seven times, those 10 consecutive windows were merged into a 60 s window. Apart from providing good frequency resolution for the vital rate computation, this merging method further solidifies range bin selection. When a range bin was selected multiple times consecutively, the overall chance of mistaken selection is lower than for each original time window.

Before running the range bin selection algorithm, we manually narrowed down the available range bins, according to the nature of the radar device and the physical dimensions of the experimental setup. We cut the range bins which were closer than 40 cm. Up to 40 cm, crosstalk between transmitter and receiver antennas occurs and corrupts the signals. On the other end, we cut off range bins further away than 3 m. While the distance to the wall right behind the bed was 2.5 m, we observed that in many cases, good signals were found in somewhat further range bins. This was caused by waves that were not directly reflected back from the participant to the radar (multipath reflections).

### 2.8. Dataset Size and Algorithm Run Time

The algorithms were implemented in the Python programming language, version *3.12.2*. They were executed on a computing server with two Intel Xeon Gold 6240 CPUs and 768 GB of RAM. The algorithm run times were recorded with the *time* package of Python.

The size of the sleep recordings from the six included participants together was 259 GB. To execute the algorithms for breathing bin selection and breathing rate computation, it took 33.5 h overall. Heartbeat range bin selection and heart rate computation took 15.5 h.

### 2.9. Comparison with Single Bin Selection

The vital sign errors of the proposed method are compared to the errors of the single bin selection method from [30] (referred to from now on as single bin selection). Both methods used 60 s time windows for vital rate computation. But the step between the start of consecutive windows was 20 s for the single bin selection and 5 s for the proposed method. Therefore, there were, overall, approximately four times as many time windows analyzed using the proposed method.

## 3. Results

### 3.1. Range Profile

Figure 7 shows the output of the proposed method for one exemplary participant. From here on, we refer to this output as the range profile. The top panel shows the range bins selected for breathing, while the bottom panel shows the ones selected for heartbeats. Yellow colors indicate the selected range bins. By comparing the two panels, the different distribution of selected range bins can be observed. Breathing range bins are selected in a wider spatial interval, and in general, further away from the radar. In other words, towards the torso of the participant. Heartbeat range bins are grouped in a narrower space, and they are located closer on average to the radar, that is, towards the feet of the participant. It is worth noting that the named source body regions assume that the participants were lying in the bed as instructed, with their feet pointed towards the radar.

### 3.2. Exploration of Windows with No Selected Range Bins

Time windows where the proposed method returned no selections were analyzed as described in Section 2.4. The results are shown in Table 1 and Table 2 for breathing and heartbeat selection, respectively. The results were computed from all six participants taken together. In terms of standard deviation, the results showed the same pattern for both breathing and heartbeat analysis. The time windows where the proposed method selected no range bins showed higher values than the time windows where at least one range bin was selected, with no overlap between the interquartile ranges of the two groups of time windows.

The results of the distribution analysis using skewness and kurtosis were as follows. In the case of breathing range bin selection, the group of windows where no bin was selected exceeded the skewness threshold three times more often (15.70% vs. 46.30%) and the kurtosis threshold four times as often (15.16% vs. 59.49%). Looking at selection based on heartbeats, the percentages are two times higher in terms of skewness (17.92% vs. 32.85%) and three times higher for kurtosis (17.76% vs. 46.92%) in the group without selection.

### 3.3. Vital Rate Computation

Breathing and heart rates were computed for all six participants (total duration 33.05 h) in time windows where the proposed method selected at least one range bin. The results from the proposed method are presented both with and without applying the chirp median.

#### 3.3.1. Breathing Rate

Table 3 presents the results of breathing rate computation. The duration where no vital rate was computed is higher for the proposed method compared to single bin selection (5 min vs. 50 min 15 s). But as the results presented in Section 3.2 showed, windows where the proposed method did not select range bins (and therefore no vital rates were computed) were corrupted by artifacts to a high degree. In terms of accuracy to the reference breathing rate, the proposed method outperformed single bin selection based on both the proportion of windows where the error was less than ±1 BPM (93.2% vs. 96.4%) and also the MAPE (2.24% vs. 1.48%).

Table 3 also shows the breathing rate results when computed without using the chirp median. The results are almost identical to those computed with the chirp median.

Figure 8 shows the breathing rate of one exemplary participant throughout the night (top panel), the error of the rates computed from radar (middle panel) and the distribution of selected range bins. The errors are generally close to 0, sporadically interrupted by higher deviations. It is noticeable how periods of higher variation in the breathing rate also coincide with changes in the range profile.

In Figure 9, the comparison of radar and reference breathing rates from all participants is presented in a Bland–Altman plot. It shows a good agreement between the reference PSG values and the radar, with a mean difference of 0.10 BPM, and the upper and lower limits of the 95% confidence interval are 0.99 and −0.80 BPM, respectively.

#### 3.3.2. Heart Rate

The results of the heart rate computation are presented in Table 4. The comparison between single bin selection and the proposed method shows a similar picture for the breathing rate results. With the proposed method, the duration for which no heart rates were computed was longer by around 2 h. But again, the windows where no rates were computed were corrupted by motion artifacts to a high degree (see Section 3.2). The proportion of windows where the radar heart rate had an error less than ±1 BPM was 20% higher with the proposed method. The MAPE using the proposed method was more than three times lower compared to single bin selection.

In contrast to the breathing rate results, the chirp median made a large difference for heart rate computation. With the chirp median, the duration without computed heart rate was lower by 2.5 h. There were 13.6% more time windows where the error was less than ±1 BPM. The MAPE was also 0.37% lower.

In addition to the general accuracy improvements, in the case of one particular participant, the benefit of the proposed method was even greater. In our previous publication, we described how the algorithm could not detect a heartbeat reliably for this participant. In contrast, with the proposed method, while heart rates could only be computed in a third of the total time windows, when they could be computed, they showed good accuracy, with a MAPE of 1.53%. Single bin selection only returned heart rates for this participant in 17% of all time windows, and had a MAPE of 31.42%, essentially guessing randomly.

Heart rates computed throughout the night for one exemplary participant are shown in Figure 10. The top panel shows the radar and reference heart rates, the middle panel shows the error of the radar heart rates, and the bottom panel shows the selected range bins. As with the breathing rate example shown in Figure 8, the connection between intervals of high variance in the heart rates and the range profile can be observed.

Figure 11 is a Bland–Altman plot comparing heart rates from all participants computed with radar and PSG. In general, there is a good agreement between the sensors, with a mean difference of −0.02 BPM, and upper and lower limits of the 95% confidence interval at 2.10 and −2.13 BPM, respectively. However, the plot shows that at heart rates below 60 BPM, the radar underestimates, while above 75 BPM, it overestimates the actual heart rate.

## 4. Discussion

In this work, a new range bin selection method for FMCW radar vital sign monitoring is presented based on persistence homology, a topological data analysis technique. This study builds upon the novel lateral radar placement strategy that we introduced in our prior publication [30]. With the new method introduced here, every range bin which recorded breathing or heartbeat signals can be selected for vital rate computation, instead of just selecting a single range bin.

### 4.1. Interpretation of Results

The first output of the multi-bin selection method is the range profile, which is the spatial distribution of selected range bins through time. In Figure 7, we presented the breathing and heartbeat range profile of one participant. The difference in the two distributions aligns well with physiological expectations. Breathing selections are centered towards the more distal part of the bed. Given that participants slept with their head away from the radar, their upper body, where breathing-related motions can be detected, would have been located in this distal part of the bed. The large number of selected bins is also coherent with the fact that the breathing motion affects the whole upper body, resulting in a large target size. In contrast, heartbeat range bins were selected closer to the radar, and are fewer. This again lines up with expectations, as the clearest heartbeats should be visible on the legs where the higher amplitude breathing motion is not present to mask them.

The analysis of time windows where no range bins were selected by the proposed method provides a view on the confidence of the algorithm. The results (see Table 1 and Table 2) indicate, based on magnitude information, that a large portion of these windows had some irregular, large motion, making them unsuitable for accurate vital rate computation. Building on this insight, in future work the multi-bin selection could be improved by incorporating signal quality computation into the selection algorithm. One such signal quality check could be based on the magnitude analysis that we introduced here. A good motivation for such improvement on the bin selection algorithm is that in sleep monitoring there is a large emphasis on having a high level of confidence in the computed rates’ validity. This means that it is better to have fewer computed vital rates in a night with high accuracy, than having vital rates throughout the night, with low accuracy.

The vital rates computed with the proposed multi-bin selection method showed an improvement in accuracy over single bin selection, both for breathing and heart rates. The improvement was especially large for the heart rates. These results support the hypothesis that by summarizing information on the same phenomenon from multiple sources, a more accurate result can be achieved.

While the new method showed an improvement in accuracy measures, in terms of the duration where no vital rates were computed, it performed worse than our previously published method. But the analysis of windows where the proposed method selected no bins suggests that the new method correctly avoided these windows. Therefore, the previous method had a higher prevalence of incorrect selections. This would also help to explain its worse vital rate performance.

We also showcased the utility of combining the individual chirps in a single frame using the chirp median. The improvement in heart rate accuracy (see Table 4) confirms our observation that the chirp median makes weak radar heartbeat signals easier to detect. Combined with the theoretical explanation of its validity presented in Section 2.2, this result provides a valuable new tool for radar heartbeat monitoring.

Compared with results presented in a review on contactless sleep vital sign monitoring [18], our proposed method delivers lower vital rate errors than most of the described methods. This is a further indication of the potential of combining the lateral radar placement with multiple range bin selection, although, it should be pointed out that most works presented in [18] tested their methods on more participants.

### 4.2. Potential Uses of Range Profile

Apart from leading to more accurate vital rate computation, the range profile that the proposed method returns could have further uses in sleep monitoring. Here, we describe ideas which could be explored in future clinical studies. The distribution of range bins in space, combined with some basic knowledge about the recording environment (such as where the radar is placed in relation to the bed and how the patient is oriented in the bed), could be used to identify body regions. The most prominent use for such identification would be the differentiation between abdomen and thorax. Comparing the breathing motion of these regions can be instrumental in the diagnosis of certain pathology such as obstructive sleep apnea syndrome (OSAS) [52] and chronic obstructive pulmonary disease (COPD) [53].

Analyzing the range profile over time could be used to detect nocturnal posture changes. This information is valuable in sleep apnea. Many patients suffering from sleep apnea have what is called positional sleep apnea, meaning that apnea events mostly, or even exclusively, occur in a certain sleeping postures (most often supine). Changes in the distribution of selected range bins could also indicate arousals from sleep, which are also markers of poor sleep quality and therefore important to detect.

Furthermore, changes in the distribution of range bins could also be indications for sleep apnea. For example, if, suddenly, no breathing signals are found in the upper body regions, but instead, heartbeat signals are suddenly detected there, it would mean a cessation of breathing movements, which leads to an unmasking of the heartbeat signals from the upper body. Alternatively, there could potentially be a different distribution of range bins showing breathing during an obstructive apnea event, as the patient is struggling to breathe against the airway obstruction.

### 4.3. Limitations and Outlook

One limitation of this work is the use of empirical criteria and thresholds in the range bin selection algorithm. While the criteria were carefully chosen, and the results validate their effectiveness, machine learning algorithms, combined with well-designed features, could potentially improve the results. Alternatively, a sensitivity analysis could be conducted to evaluate the effect of different criteria and thresholds on the performance of the proposed method.

The heart rates computed with the proposed method showed an improvement over single bin selection and also compare well with results from the literature (see comparison with [18] in Section 4.1). However, the heart rate dependent bias of the algorithm displayed in Figure 11 warrants further investigation. Future works should study whether the bias is caused by the implemented heart rate computing algorithm or the radar device itself.

Furthermore, our work is also limited by the low number of participants. This study was meant to show the feasibility of applying multiple range bin selection to achieve better vital sign monitoring performance using radar. Therefore, the proposed method was first evaluated in this study on this limited group of six participants. In the future, these algorithms should be tested on a larger group of participants, with more variation in age, sex and health status. Currently, we are conducting measurements in a clinical study, which will be suitable to evaluate the generalizability of the proposed method. The study will include a larger group of participants, with different demographic and health statuses from the participants in this work.

This work showed the potential of using persistence diagram-based algorithms to improve radar vital sign monitoring during sleep. Further studies could investigate implementing these algorithms on less computationally powerful systems to facilitate widespread adoption. Optimization techniques like point cloud subsampling before Vietoris–Rips filtration and parallelizing computation on the individual range bins are two possible candidates for making the proposed method less computationally expensive and possibly suitable for real-time applications.

Other potential next steps would be to test the possible uses of the range profile that were listed above. With an expert-annotated clinical sleep dataset, these could be properly evaluated. Furthermore, as already mentioned, the development of a signal quality indicator algorithm has the potential to further improve the validity of radar-based sleep monitoring.

## 5. Conclusions

In summary, in this publication, we presented the feasibility of computing more accurate vital rates from an FMCW radar by using multiple range bin selection instead of single bin selection. For this, we used an algorithm based on modern time series analysis techniques from the field of topological data analysis. In addition to the derived vital rates, we presented the distribution of selected range bins as a potential information source in itself. Furthermore, we showed a method called chirp median, which can improve the heartbeat detection using FMCW radar.

## Figures and Tables

**Figure 1 sensors-25-02596-f001:**
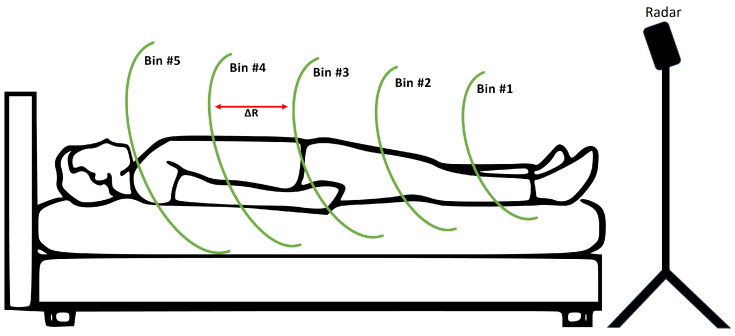
Schematic representation of the radar range bins’ distribution along the human body. The radar device is located at the foot end of the bed on a tripod. Range bin borders are represented with green lines. The distance between adjacent range bin borders ΔR is 5 cm (figure not to scale). This cartoon example shows how signals can be recorded separately from sections of the legs (bins #1–#3), the abdomen (bin #4) and the thorax (bin #5).

**Figure 2 sensors-25-02596-f002:**
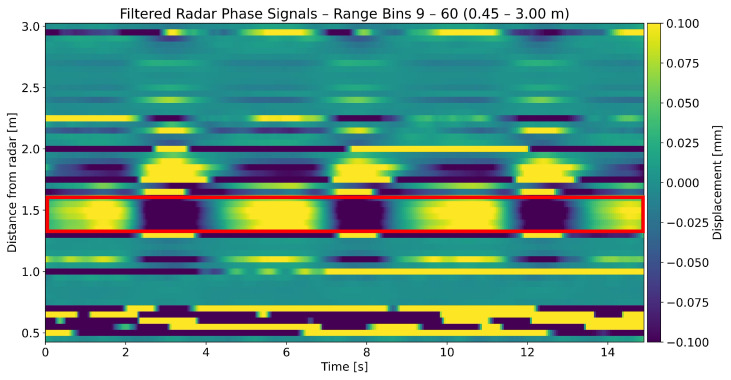
A 15 s time window showing the displacement in multiple range bins, estimated from the radar phase signal. The displacement values are represented by the coloring (see colorbar). One row of the figure corresponds to one range bin. The red framing shows a group of range bins which coherently display a clear breathing motion. The breathing motion can be seen by the periodic change of light and dark colors, which correspond to high and low displacement values, respectively. To make the breathing motion visible in the lower amplitude range bins as well, the displacement values are cut off above 0.1 mm and below −0.1 mm.

**Figure 3 sensors-25-02596-f003:**
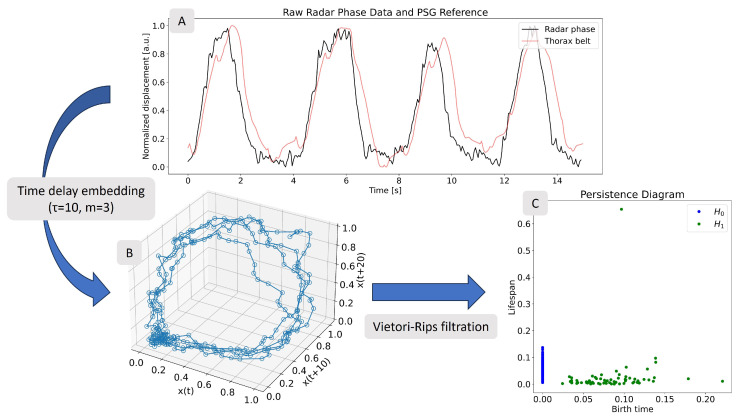
Process of going from time series (**A**) to persistence diagram (**C**) using Vietoris–Rips filtration. Before the filtration can be applied, the in-between step of time delay embedding is performed using embedding parameters τ = 10 and m = 3 (**B**). The embedding shows a loop which is characteristic for periodic signals. The persistence diagram computed with Vietoris–Rips filtration has a single one-dimensional feature with high lifetime, corresponding to the loop of the embedding.

**Figure 4 sensors-25-02596-f004:**
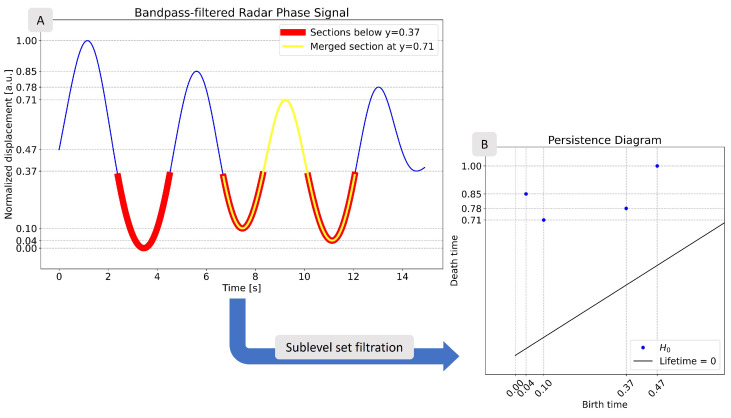
Example for sublevel set filtration. (**A**) shows the input time series. For the sake of a clear visualization, a band-pass filter has been applied on the signal. By computing the sublevel set filtration, we obtain the persistence diagram presented in (**B**). The grid lines in the plots show which parts of the time series are represented by the points of the persistence diagram.

**Figure 5 sensors-25-02596-f005:**
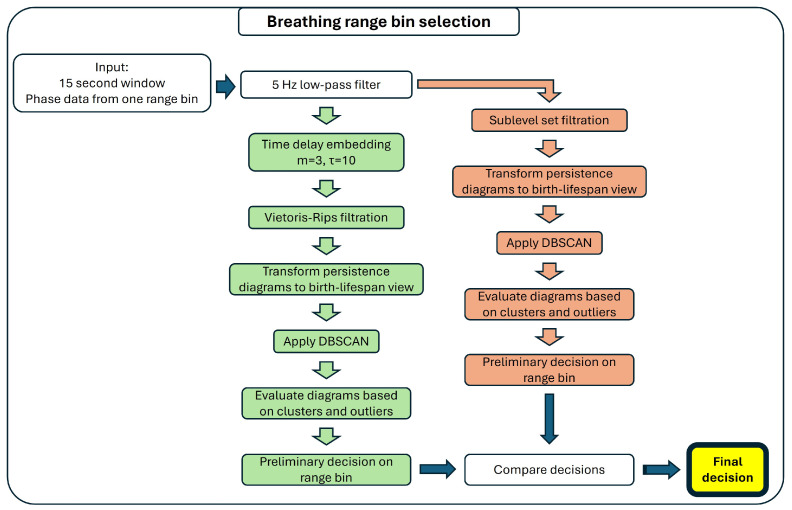
Flowchart of the breathing bin selection algorithm. The steps for the Vietoris–Rips filtration-based processing are highlighted with green, while the steps for the sublevel set filtration-based processing are highlighted with red.

**Figure 6 sensors-25-02596-f006:**
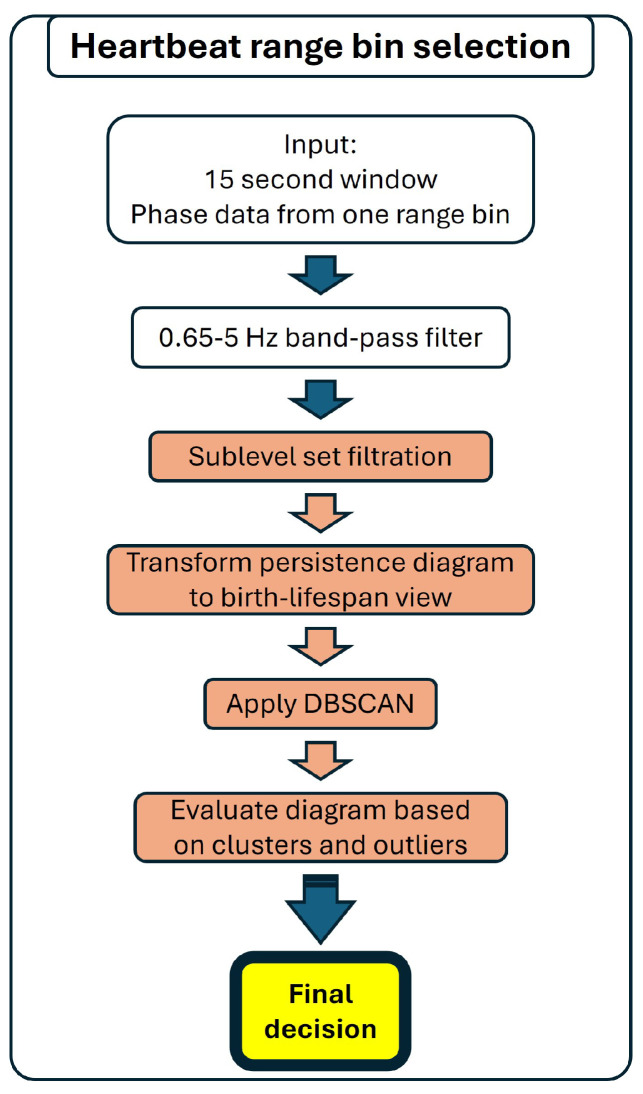
Flowchart of the heartbeat bin selection algorithm. The steps for the sublevel set filtration-based processing are highlighted with red.

**Figure 7 sensors-25-02596-f007:**
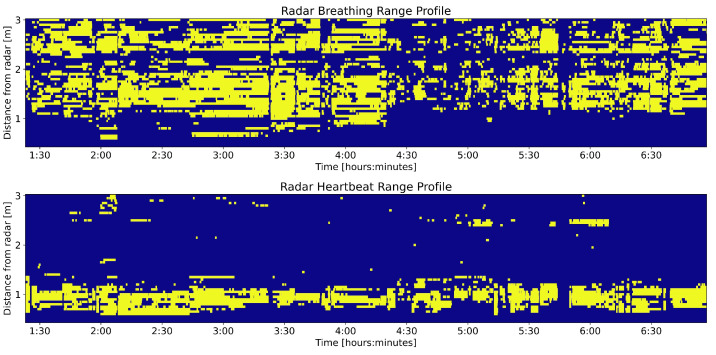
Range profile of one participant for breathing and heartbeat activity. The horizontal axis indicates the time from recording start, while the vertical axis shows the distance from the radar device (bottom is closer, top is further away). Yellow color indicates selected range bins for breathing (**top** panel) and heartbeat (**bottom** panel).

**Figure 8 sensors-25-02596-f008:**
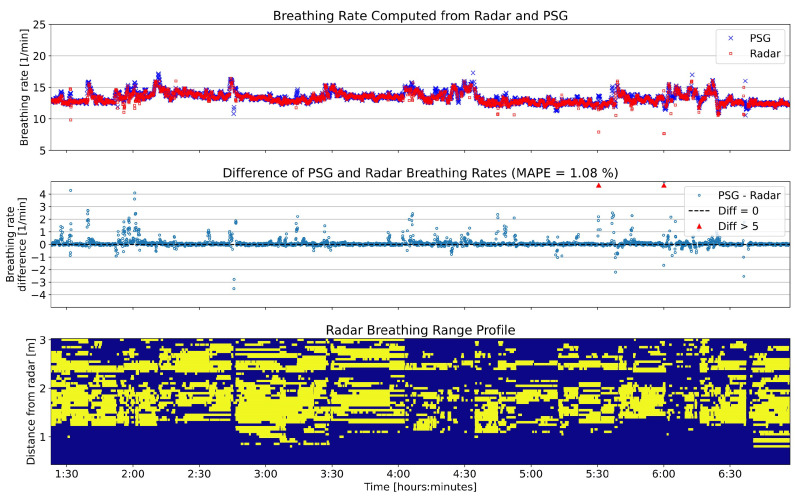
Breathing rate of one participant throughout the night; comparison between radar and PSG. The (**top**) panel shows the computed breathing rates. The (**middle**) panel shows the difference between the two sensors’ rates throughout the night and a summary statistic with the mean absolute percent error (MAPE). The (**bottom**) panel is the range profile as already presented in Figure 7. The *x*-axis is shared between the three panels; therefore, to reduce visual clutter, the labels are only displayed for the (**bottom**) panel.

**Figure 9 sensors-25-02596-f009:**
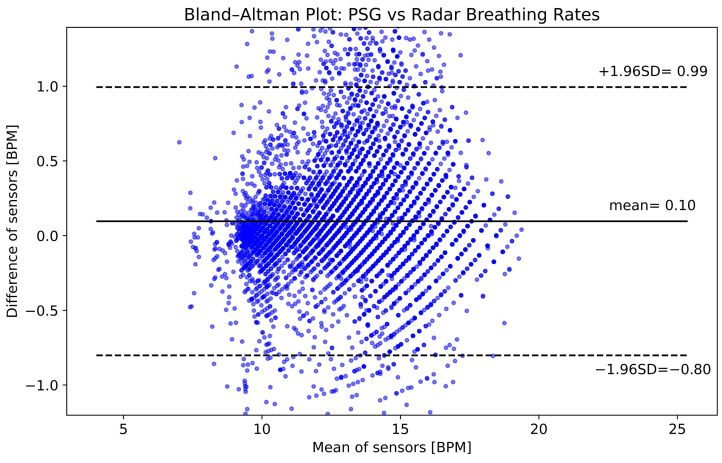
Bland–Altman plot comparing the breathing rates computed using PSG and radar data from all six participants. The horizontal axis shows the average of the two rates, while the vertical axis shows their difference, both in units of breaths per minute (BPM).

**Figure 10 sensors-25-02596-f010:**
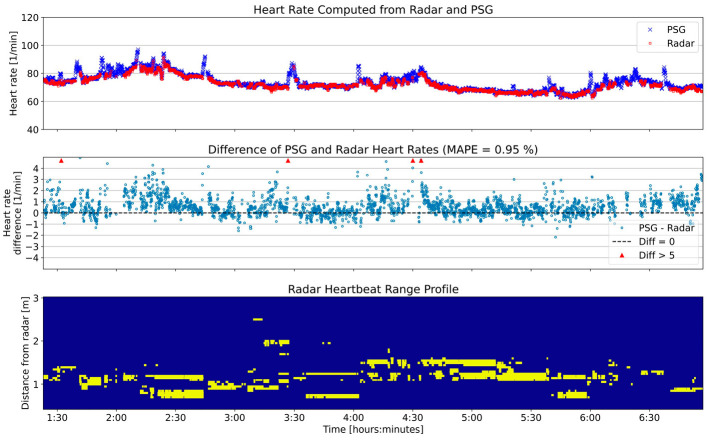
Heart rate of one participant (same participant as in Figure 8) throughout the night; comparison between radar and PSG. The (**top**) panel shows the computed heart rates. The (**middle**) panel shows the difference between the two sensors’ rates throughout the night and a summary statistic with the mean absolute percent error (MAPE). The (**bottom**) panel is the range profile as already presented in Figure 7. The x-axis is shared between the three panels; therefore to reduce visual clutter, the labels are only displayed for the (**bottom**) panel.

**Figure 11 sensors-25-02596-f011:**
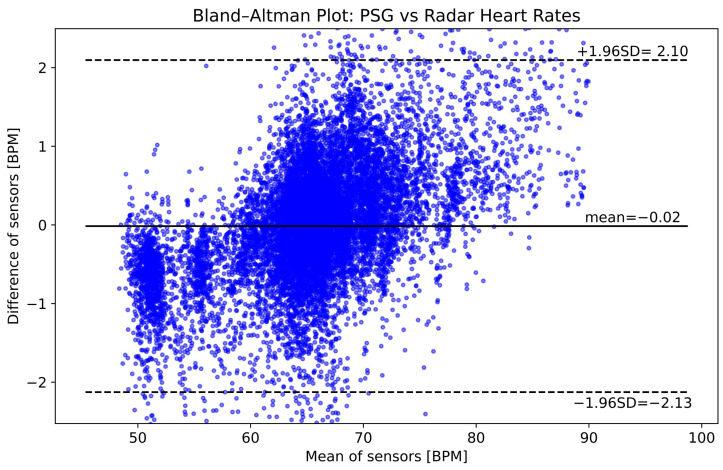
Bland–Altman plot comparing heart rates computed using PSG and radar data from all six participants. The horizontal axis shows the average of the two rates, while the vertical axis shows their difference, both in units of beats per minute (BPM).

**Table 1 sensors-25-02596-t001:** Exploration of windows where no range bins were selected to detect breathing signals.

	Time Windows Where at Least One Range Bin Was Selected for Breathing	Time Windows Where No Range Bins Were Selected for Breathing
Standard deviation quartiles [V]	170.06–240.93–329.88	650.44–879.18–1140.04
Percentage of windows where abs. skewness > 1 [%]	15.70	46.30
Percentage of windows where abs. kurtosis > 3 [%]	15.16	59.49

**Table 2 sensors-25-02596-t002:** Exploration of windows where no range bins were selected to detect heartbeat signals.

	Time Windows Where at Least One Range Bin Selected Was for Heartbeat	Time Windows Where No Range Bins Were Selected for Heartbeat
Magnitude standard deviation quartiles [V]	50.47–98.92–192.85	280.83–473.13–735.26
Percentage of windows where abs. skewness > 1 [%]	17.92	32.85
Percentage of windows where abs. kurtosis > 3 [%]	17.76	46.92

**Table 3 sensors-25-02596-t003:** Breathing rate results. Comparison between single bin selection and proposed method (with and without chirp median).

	Single Bin Selection [30]	Proposed Method with Chirp Median	Proposed Method Using 1st Chirp
Windows where PSG rate was computed [% (count)]	94 (5583)	93.6 (22,563)	93.6 (22,563)
Windows where radar rate was computed [% (count)]	98.5 (5851)	90 (21,680)	89.8 (21,651)
Duration where no PSG rate was computed [hh:mm:ss]	00:12:00	00:04:25	00:04:25
Duration where no radar rate was computed [hh:mm:ss]	00:05:00	00:50:15	00:52:15
Windows where difference between PSG and radar < ± 1 BPM [% (count)]	93.2 (5151)	96.4 (20,182)	96.4 (20,167)
Mean Absolute Error [1/min]	0.29	0.20	0.20
Mean Absolute Percent Error [%]	2.24	1.48	1.50

BPM: breaths per minute.

**Table 4 sensors-25-02596-t004:** Heart rate results. Comparison between single bin selection and proposed method (with and without chirp median).

	Single Bin Selection [30]	Proposed Method with Chirp Median	Proposed Method Using 1st Chirp
Windows where PSG rate was computed [% (count)]	92.6 (5503)	92.4 (22,279)	92.4 (22,279)
Windows where radar rate was computed [% (count)]	70.6 (4194)	59.7 (14,379)	54.4 (13,101)
Duration where no PSG rate was computed [hh:mm:ss]	01:48:00	01:41:00	01:41:00
Duration where no radar rate was computed [hh:mm:ss]	05:53:40	07:45:55	10:10:35
Windows where difference between PSG and radar < ± 1 BPM [% (count)]	58.2 (2332)	81.6 (11,379)	68.0 (8628)
Mean Absolute Error [1/min]	1.97	0.66	0.93
Mean Absolute Percent Error [%]	3.25	1.02	1.39

BPM: beats per minute.

## Data Availability

Data sharing is not applicable due to concerns for the human participants’ privacy. The source code of this article is available at the following repository: https://doi.org/10.5281/zenodo.14926278.

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
