# Peer review of "Feasibility of Radar Vital Sign Monitoring Using Multiple Range Bin Selection"

_sensors, 2025, doi:10.3390/s25082596_

Round 1

Reviewer 1 Report

Comments and Suggestions for Authors

This paper presents a range bin selection method for FMCW radar vital sign monitoring based on persistence homology. Experiment results illustrate that more accurate vital rates can be computed than using single bin selection. There are some issues need to be considered.

  1. Firstly, the description of the proposed multiple range bin selection method is not clear enough. How is the persistence diagram obtained? What is the difference of detection methods between breathing and heartbeat? It is recommended to provide flow diagram or formula description to make the method clearer.

  1. What is the mathematical or physical meaning of the time delay embedding? What is the effect of different embedding parameters on the time delay embedding figure?

  1. As shown in Figure 2, the displacement of estimated from radar phase signals is less than 0.1mm. However, it is mentioned that the range accuracy of the phase signal in one range bin is below 1 mm in Section 2.1. It seems that the range accuracy is worse than the displacement. Then, how to obtain an accurate estimation of the displacement?

  1. In the schematic representation diagram shown in Figure 1, the human body is in a lying position. If the body is in different lying positions such as on the right side or left side, are accurate measurements still available?

Reviewer 2 Report

Comments and Suggestions for Authors

Comments for sensors-3526129-peer-review-v1

In this manuscript, there are some significance and novelty as follows.

  1. It proposes a multi-range unit selection algorithm based on persistence graph for non-contact vital signs monitoring using FMCW radar. This method improves on the existing single-range unit selection method by being able to extract respiratory and heartbeat signals from multiple body parts at the same time, which improves the robustness and accuracy of vital signs monitoring.
  2. The article applies the persistent homotopy algorithm in topological data analysis (TDA) to time series analysis, especially for the detection of non-smooth biomedical signals. This approach is novel in the field of biomedical signal processing and demonstrates the potential of TDA in processing complex biomedical data.
  3. The study demonstrates the promising application of the method in sleep monitoring, in particular by improving the accuracy of respiration and heartbeat monitoring through multi-range cell selection. This is important for multiple applications in sleep medicine, such as the diagnosis of sleep apnea syndrome.

Therefore, there are some potential reasons for acceptance as follows.

  1. The experimental results show a significant improvement in the accuracy of respiration and heartbeat monitoring using a multi-range unit selection method compared to a single-range unit selection method. This result has high clinical application value and can significantly improve the effectiveness of non-contact vital signs monitoring.
  2. The multi-range unit selection algorithm based on persistence graph proposed in the article is highly innovative and demonstrates a high level of complexity and depth in its technical implementation, reflecting the authors' expertise in the field.
  3. The method not only has the potential to be applied in sleep monitoring, but also can be extended to other fields that require non-contact vital signs monitoring, such as telemedicine, elderly care, etc. It has a wide range of application prospects.

However, there are still some potential reasons for rejection as follows.

  1. The study was validated using data from only 6 healthy participants, which is a very small sample size and may not be sufficient to fully validate the effectiveness and robustness of the method. Data from larger clinical trials may be more convincing.
  2. Although the method has improved in accuracy, its complexity and computational cost may be high, and the practical application may face the problems of computational resources and real-time performance, which need to be further optimized and validated.
  3. The article mainly compares the effects of single-range unit selection and multi-range unit selection methods, but lacks comparative experiments with other state-of-the-art methods, which can not fully demonstrate the relative advantages of the method.

In addition, there are some other suggestions for improvement as follows.

  1. It is recommended that the effectiveness and robustness of the method be verified in larger-scale clinical trials, especially including participants of different ages, genders, and health statuses, in order to improve the generalizability and credibility of the results.
  2. Further optimize the algorithms to reduce computational complexity and cost, and ensure real-time performance and operability in practical applications. At the same time, hardware acceleration techniques, such as GPU computing, can be considered to increase processing speed.
  3. Introduce more comparative experiments, especially with other advanced non-contact vital signs monitoring methods, to comprehensively demonstrate the advantages and shortcomings of the method and enhance the persuasiveness and competitiveness of the research.
Comments on the Quality of English Language

No

Round 2

Reviewer 1 Report

Comments and Suggestions for Authors

The previous concerns have been addressed in the revised manuscript. More details are added which make the paper clearer. The quality of the paper has been improved.

Reviewer 2 Report

Comments and Suggestions for Authors

Review of the Manuscript: "Feasibility of Radar Vital Sign Monitoring using Multiple Range Bin

Selection" (ID: sensors-3526129-peer-review-v2)

Overall Assessment

The manuscript presents a novel method for improving radar-based vital sign monitoring using topological data analysis (TDA) and multiple range bin selection. The study is well-structured, methodologically sound, and addresses a relevant gap in contactless sleep monitoring. Below are the detailed strengths and weaknesses, followed by a recommendation.

Strengths

  1. Innovative Approach:

   - The use of persistence diagrams from TDA for radar signal analysis is original and aligns well with the challenges of non-stationary vital sign signals. 

   - The lateral radar placement strategy and multiple bin selection improve robustness compared to single-bin methods. 

  1. Technical Rigor:

   - Detailed explanation of FMCW radar principles, signal processing (e.g., chirp median), and TDA (Vietoris-Rips/sublevel set filtrations) demonstrates strong technical foundations. 

   - The DBSCAN-based criteria for bin selection are well-justified and empirically validated. 

  1. Experimental Validation:

   - Comparative results against single-bin selection show clear improvements in accuracy (e.g., heart rate MAE reduced from 1.97 to 0.66 BPM). 

   - The analysis of unselected windows (high skewness/kurtosis) supports the method’s ability to reject artifact-corrupted data. 

  1. Clinical Relevance:

   - The proposed method could supplement polysomnography (PSG) by reducing sensor intrusiveness. The range profile’s potential for detecting body position changes and sleep apnea is compelling. 

  1. Clarity and Organization:

   - Figures (e.g., range profiles, Bland-Altman plots) effectively illustrate key results. 

   - The manuscript follows a logical flow, with clear subsections for methods, results, and discussion. 

Weaknesses and Suggestions for Improvement. 

  1. Limited Dataset:

   - Only 6 healthy participants were included, raising concerns about generalizability. The authors acknowledge this but should emphasize plans for larger/clinical validation.  

   - Suggestion: Add a power analysis or justification for sample size. 

  1. Empirical Thresholds:

   - Some criteria (e.g., DBSCAN parameters, chirp median thresholds) were empirically set. While results validate these choices, a sensitivity analysis or machine-learning alternative could strengthen robustness. 

  1. Computational Efficiency:

   - The algorithm runtime (33.5 hours for breathing analysis) may limit real-time applicability. 

   - Suggestion: Discuss potential optimizations (e.g., parallel processing, hardware acceleration). 

  1. Heart Rate Bias:

   - The Bland-Altman plot (Figure 11) shows underestimation at low heart rates and overestimation at high rates. This systematic error warrants further investigation. 

  1. Minor Issues:

   - Typographical errors (e.g., "mulitmodal" → "multimodal," "electrcardiogram" → "electrocardiogram"). 

   - Figure 2’s colorbar is mentioned but not visible in the excerpt. Ensure all figures are properly labeled. 

Recommendation 

The manuscript is ‘acceptable for publication’ after minor revisions addressing the above points. The study advances radar-based vital sign monitoring with a rigorous and innovative methodology. Its clinical potential, particularly for sleep studies, justifies publication pending: 

  1. Clarification of empirical thresholds.
  2. Discussion of computational scalability.
  3. Proofreading for minor errors.

Rating: Accept with minor revisions 

Additional Notes: 

- The authors should ensure all abbreviations (e.g., OSAS, COPD) are defined at first use. 

- Consider adding a supplementary video demonstrating the range profile’s temporal dynamics. 

- Future work could explore integration with other modalities (e.g., infrared cameras) for multi-signal validation. 

Overall, this work contributes meaningfully to the field of contactless health monitoring and aligns well with the journal’s scope.
